# HMGB1-Mediated Cell Death—A Crucial Element in Post-Hepatectomy Liver Failure

**DOI:** 10.3390/ijms25137150

**Published:** 2024-06-28

**Authors:** Laura Brunnthaler, Thomas G. Hammond, David Pereyra, Jonas Santol, Joel Probst, Valerie Laferl, Ulrike Resch, Monika Aiad, Anna Sofie Janoschek, Thomas Gruenberger, Hubert Hackl, Patrick Starlinger, Alice Assinger

**Affiliations:** 1Department of Vascular Biology and Thrombosis Research, Centre of Physiology and Pharmacology, Medical University of Vienna, Schwarzspanierstrasse 17, 1090 Vienna, Austria; laura.brunnthaler@meduniwien.ac.at (L.B.); tellsantol@gmail.com (J.S.); ulrike.resch@meduniwien.ac.at (U.R.); 2Division of Molecular and Systems Toxicology, Department of Pharmaceutical Sciences, University of Basel, 4055 Basel, Switzerland; thomas.hammond@astrazeneca.com; 3Clinical Pharmacology and Safety Sciences, AstraZeneca, Cambridge CB4 0WG, UK; 4Department of General Surgery, Division of Visceral Surgery, Medical University of Vienna, General Hospital, 1090 Vienna, Austria; david.pereyra@meduniwien.ac.at (D.P.); telllaferl@gmail.com (V.L.); tellaiad@gmail.com (M.A.); telljanoschek@gmail.com (A.S.J.); 5Department of Surgery, HPB Center, Viennese Health Network, Clinic Favoriten and Sigmund Freud Private University, 1100 Vienna, Austria; tellprobst@gmail.com (J.P.); tgruenberger@icloud.com (T.G.); 6Department of Surgery, Division of Hepatobiliary and Pancreas Surgery, Mayo Clinic, Rochester, MN 55905, USA; 7Institute of Bioinformatics, Biocenter, Medical University of Innsbruck, 6020 Innsbruck, Austria; hubert.hackl@i-med.ac.at

**Keywords:** partial hepatectomy, post hepatectomy liver failure, HMGB1, caspase-cleaved keratin-18

## Abstract

Liver resection (LR) is the primary treatment for hepatic tumors, yet posthepatectomy liver failure (PHLF) remains a significant concern. While the precise etiology of PHLF remains elusive, dysregulated inflammatory processes are pivotal. Therefore, we explored the theragnostic potential of extracellular high-mobility-group-box protein 1 (HMGB1), a key damage-associated molecular pattern (DAMP) released by hepatocytes, in liver recovery post LR in patients and animal models. Plasma from 96 LR patients and liver tissues from a subset of 24 LR patients were analyzed for HMGB1 levels, and associations with PHLF and liver injury markers were assessed. In a murine LR model, the HMGB1 inhibitor glycyrrhizin, was administered to assess its impact on liver regeneration. Furthermore, plasma levels of keratin-18 (K18) and cleaved cytokeratin-18 (ccK18) were quantified to assess suitability as predictive biomarkers for PHLF. Patients experiencing PHLF exhibited elevated levels of intrahepatic and circulating HMGB1, correlating with markers of liver injury. In a murine LR model, inhibition of HMGB1 improved liver function, reduced steatosis, enhanced regeneration and decreased hepatic cell death. Elevated levels of hepatic cell death markers K18 and ccK18 were detected in patients with PHLF and correlations with levels of circulating HMGB1 was observed. Our study underscores the therapeutic and predictive potential of HMGB1 in PHLF mitigation. Elevated HMGB1, K18, and ccK18 levels correlate with patient outcomes, highlighting their predictive significance. Targeting HMGB1 enhances liver regeneration in murine LR models, emphasizing its role in potential intervention and prediction strategies for liver surgery.

## 1. Introduction

Liver resection (LR) is the primary therapeutic approach for individuals diagnosed with hepatic tumors, owing to the liver’s remarkable regenerative capacity [1]. However, insufficient hepatic regeneration can lead to post hepatectomy liver failure (PHLF), which despite notable advances in surgical techniques, remains a significant clinical concern associated with patient morbidity and mortality. Various methods are available to assess PHLF, including classical liver volumetry and the invasive hepatic venous pressure gradient (HVPG). Alternatively, non-invasive tools such as indocyanine green (ICG) clearance, LiMAx, and the recently established APRI+ALBI score are increasingly used to predict and assess PHLF [2,3,4].

Once PHLF occurs curative options are limited, undermining the critical need for new therapeutic interventions to improve post-surgical liver function.

Inflammatory stimulation is key to initiate hepatic regeneration processes. However, excessive inflammation impedes liver repair [5,6]. A central molecule released in response to injury is high-mobility group box protein-1 (HMGB1) [7], a versatile nonhistone nuclear protein with various roles depending on its cellular localization. Within the nucleus, it acts as a DNA chaperone, maintaining chromosome structure and function, while cytoplasmatic HMGB1 can induce autophagy. Upon secretion, extracellular HMGB1 functions as a damage-associated molecular pattern (DAMP) molecule, modulating inflammation and immune responses [8].

In inflammatory scenarios, extracellular HMGB1 primarily originates from hepatocytes [9] and assumes diverse functions depending on the cellular or disease context. This versatility renders it a central mediator in various liver diseases [10]. Although HMGB1 plays a pivotal role in acute liver injury, liver-specific HMGB1 deficiency does not affect chronic injury responses, such as fibrosis, regeneration, and inflammation [11]. As of now, the role of HMGB1 in patients undergoing LR remains unknown, and murine data are inconclusive, with no mechanistic role of HMGB1 being elucidated. In animal models of LR, hepatic HMGB1 knockout did not impact hepatocyte proliferation [11], while in wildtype mice systemic inhibition of HMGB1 enhanced liver regeneration after LR [12].

Hence, we aimed to uncover the role of HMGB1 in patients, exploring its potential as a theragnostic tool for PHLF, and delve into the underlying mechanisms of HMGB1 in liver regeneration. To achieve this, we analyzed a large patient cohort undergoing LR and utilized a mouse model of LR with underlying liver inflammation to mimic the human scenario.

## 2. Results

### 2.1. Patient Characteristics

A total of 96 LR patients (PHLF = 22) were included and analyzed for plasma parameters, with 24 patients (PHLF = 10) undergoing liver tissue analysis. Summary of patient characteristics for both cohorts can be found in Table 1 (plasma) and Table 2 (staining).

### 2.2. HMGB1 in the Regenerating Liver Is Elevated in Patients That Develop PHLF

Analysis of liver tissues obtained from PreOP and PostOP LR revealed that patients developing PHLF exhibited higher mean fluorescence levels (MFI) of HMGB1 PostOP compared to patients with functional liver regeneration (Figure 1A). Interestingly, PHLF patients appeared to have increased extracellular HMGB1 levels compared to noPHLF patients, which show a more nucleus-concentrated pattern of HMGB1. This observation was supported by plasma analysis of HMGB1, revealing that PHLF patients had higher circulating HMGB1 levels on POD1 compared to noPHLF patients (Figure 1B). The diagnostic performance of HMGB1 in plasma measured by receiver operator characteristic (ROC) analysis estimated an area under the curve (AUC) of 0.62 for PreOP, 0.68 for POD1 and 0.62 for POD5 (Figure 1C). Furthermore, in PHLF patients correlations were observed between plasma levels of HMGB1 and liver injury markers AST and ALT (R = 0.55 and R = 0.62; Figure 1D).

### 2.3. Inhibition of HMGB1 Improves Proliferation and Reduces Apoptosis in a LR Mouse Model

To explore the therapeutic potential of HMGB1 inhibition in liver regeneration, we administered glycyrrhizin, a potent HMGB1 inhibitor, to a LR model of metabolically challenged mice (Appendix A), which led to a significant decrease in HMGB1 levels (Appendix A). Improvement in liver function and architecture was evident through measurements of ALT and AST, as well as H&E and oil red O staining (Figure 2A–C). Furthermore, increased proliferation, indicative of improved liver regeneration, was observed through gene expression analysis of KI67, PCNA, and Cyclin D1 via qPCR, as well as KI67 staining (Figure 2D,E). Moreover, a reduction in apoptosis was observed by decreased expression of p21 and cleaved caspase 3 (cCasp3), following HMGB1 inhibition (Figure 2F and Appendix A).

### 2.4. Cell Death Marker ccK-18 Is Associated with HMGB1 and PHLF Development in Patients

Plasma analysis of total K18 showed elevated levels at POD1 compared to other time points. Additionally, ccK18 plasma levels PreOP and POD5 were higher in PHLF patients compared to noPHLF patients (Figure 3A). ROC analysis showed that the AUC for K18 was 0.61, while the AUC for ccK18 was 0.52 (Figure 3B) and no synergistic effects were observed, when the markers were combined with HMGB1, (Figure 3B). Furthermore, correlations between total K18, ccK18, and HMGB1 were observed, further indicating their involvement in liver regeneration (Figure 3C). We further analyzed correlations between HMGB1 and markers of neutrophil activation, including neutrophil elastase (NE), myeloperoxidase (MPO), and citrullinated histone 3 (citH3), but found no associations (Appendix A). To compare the performance of HMGB1 with established predictive markers of PHLF, we analyzed ROC curves of HMGB1 against PDR and R15 (Figure 3D) as well as ABRI+ALBI (Figure 3E). However, this analysis was limited to a subgroup due to the availability of ICG clearance data for 64 patients and ABRI+ALBI data for only 39 patients.

## 3. Discussion

Our study unveiled first human evidence that circulating and intrahepatic HMGB1 levels were elevated in patients who developed PHLF after LR, suggesting a possible association between HMGB1 and the pathogenesis of PHLF. Our murine LR model further demonstrated that inhibition of HMGB1 improved liver function, reduced steatosis, enhanced regeneration, and decreased hepatic cell death, highlighting the therapeutic potential of targeting HMGB1 in mitigating liver injury and promoting regeneration post LR. Moreover, our findings implicate HMGB1 in the regulation of hepatic cell death pathways, as evidenced by elevated levels of cell death markers K18 and ccK18 in PHLF patients. This suggests a potential mechanistic link between HMGB1-mediated cell death and the development of PHLF, underscoring the multifaceted role of HMGB1 in liver pathophysiology. In patients without PHLF, keratin 18 significantly increased on POD1 and decreased again on POD5. In contrast, patients with PHLF showed a trend toward higher keratin 18 plasma levels, but no significant changes were observed before surgery and up to five days post-surgery. The apoptotic marker ccK18 was significantly elevated in PHLF patients compared to those without PHLF before surgery, suggesting underlying liver disease in these individuals. When we analyzed the diagnostic performance of these markers, plasma levels of HMGB1 before as well as after surgery were predictive for HMBG1, but no synergistic effects were observed when HMGB1 was combined with K18 and ccK18.

Notably, intrahepatic HMGB1 translocation occurs very early after liver resection, suggesting that it might be associated with liver-specific alterations even before surgery, rather than complications arising from the surgery itself.

Plasma levels of HMGB1 could aid in identifying patients at risk of developing PHLF and implementing appropriate preventive measures or interventions. When we compared the predictive performance with established predictors of PHLF, HMGB1 showed almost comparable performance with parameters of ICG clearance but not as efficient as ABRI+ALBI. ICG-clearance is recognized as the gold standard for evaluating liver function before hepatic resection, showing associations with PHLF and postoperative mortality. However, its accuracy can be affected by factors like liver perfusion, portal flow alterations, and cholestasis, which complicate interpretation in certain patient populations. Since data were only available in a subset of patients, caution is advised in interpreting the findings, and further studies are needed to validate the predictive value of plasma HMGB1. Large-scale studies assessing various tumor types or underlying liver diseases are necessary to determine whether HMGB1 alone or in combination with other parameters could serve as a valuable tool for preoperative risk assessment.

The role of HMGB1 in liver regeneration is currently under debate as previous research has presented conflicting views on the function of HMGB1 in regenerative processes. While HMGB1 serves as important mediator to accelerate local liver damage and systemic inflammation during the early stage of metabolic dysfunction-associated steatotic liver diseases (MASLD), it appears to coordinate tissue repair and inflammation by switching among alternative redox forms as fully reduced HMGB1 expedites repair processes by interacting with CXCR4 to induce immune cell recruitment [13]. In our study as well as in previous reports no correlation between HMGB1 and neutrophil recruitment could be observed [11] arguing against reduced HMGB1-mediated repair processes in this disease context. During liver regeneration, hepatocytes undergo rapid proliferation, which leads to increased metabolic activity and elevated levels of reactive oxygen species generation, which likely render HMGB1 into a pro-inflammatory state as postulated for LPS-mediated inflammation [10]. HMGB1 exacerbates inflammation through activation of TLR4/MyD88 signaling and RAGE receptors in hepatocytes, resulting in pyroptosis, liver damage, steatosis, and impaired liver function [14,15,16]. Additionally, HMGB1 influences mitochondrial oxidative phosphorylation, free fatty acid ß-oxidation, and immune cell recruitment and activation, highlighting its multifaceted involvement in liver disease progression [17,18]. Notably, our study demonstrates that inhibiting HMGB1 with glycyrrhizin promotes heightened liver cell proliferation and enhances liver architecture while simultaneously mitigating liver injury, cell cycle arrest, and apoptosis, ultimately resulting in augmented liver regrowth.

Glycyrrhizin, which directly binds to HMGB1, rendering it unavailable for interaction with its receptors, has already been investigated in clinical trials including in autoimmune liver diseases. Moreover, various humanized anti-HMGB1 monoclonal antibodies and HMGB1-neutralizing molecules have been developed and investigated in preclinical studies on inflammatory conditions, showing promise for clinical translation.

In summary, our study highlights the potential therapeutic and predictive potential of HMGB1 in PHLF. Elevated levels of plasma HMGB1 were linked to patient outcomes, underscoring their predictive significance. Targeting HMGB1 improved liver function and regeneration in murine LR models, emphasizing its role in PHLF and offers insights for improving LR outcomes and lowering PHLF incidence.

## 4. Methods

### 4.1. Patient Cohort

Plasma samples from 96 LR patients (January 2013 to July 2017) at the General Hospital of Vienna and Clinic Landstrasse, Vienna were analyzed. Blood was taken pre-surgery (PreOP), one (POD1) and five (POD5) days after surgery. Demographics, tumor type, pre-op factors, morbidity, and PHLF (using ISGLS) were assessed [19]. All patients with liver cirrhosis were Child-Pugh class A without encephalopathy, with compensated liver function. Informed consent was obtained, adhering to ethical guidelines, and registered on ClinicalTrials.gov (NCT01921985, NCT01700231; accessed on 22 January 2024). In a subset of 24 LR patients, tissue samples were taken before and 2 h after portal branch ligation to study liver regeneration. Morbidity was defined using the criteria put forth by Dindo et al. [20]. Severe morbidity was classified as grade 3 or higher. PHLF was defined and graded according to the criteria established by the International Study Group of Liver Surgery. The development of PHLF was characterized by elevated serum bilirubin (SB) levels and prolonged prothrombin time (PT) persisting on the fifth postoperative day. If SB and PT were already abnormal before surgery, SB had to be higher and PT lower than the preoperative values. Patients who were excluded from routine postoperative blood draws due to good clinical performance or early discharge were categorized as not having PHLF. To accurately represent the proportion of patients with clinically significant, symptomatic PHLF, it was classified as PHLF grades B and C (PHLF B + C), while the absence of PHLF was indicated by no PHLF or PHLF grade A. In a subset of patients (*n* = 64) data from indocyanine green (ICG) clearance was available. This was measured in patients receiving 0.25 mg/kg ICG intravenously on the day before the liver resection. The plasma disappearance rate (PDR) and the ICG retention rate (R15) were measured by pulse spectrometry with a LiMON device (Pulsion Medical Systems, Munich, Germany).

### 4.2. Plasma Analysis

Whole blood samples were collected into pre-chilled CTAD (citrate, theophylline, adenosine, and dipyridamole) tubes, immediately placed on ice, and processed within 30 min. Blood samples were then centrifuged at 1000× *g* at 4 °C for 10 min and the supernatants collected and centrifuged at 10,000× *g* at 4 °C for another 10 min to remove any remaining platelets. Plasma samples were aliquoted and stored at −80 °C until analysis. HMGB1 levels were measured by ELISA (Shino-Test HMGB1 ELISA kit, ST51011, IBL International, Mannedorf, Switzerland). Caspase-cleaved cytokeratin-18 (ccK18) and total keratin-18 (K18) were quantified using the M30 Apoptosense^®^ ELISA kit and the M65^®^ Classic ELISA kit (both Peviva^®^, TecoMedical, Sissach, Switzerland), respectively. Serum alanine-aminotransferase (ALT) or aspartate-aminotransferase (AST) and other clinical chemistry parameters were analyzed by the hospital.

### 4.3. Animal Treatments

Animal experiments were approved by the Austrian Federal Ministry for Science, Research and Economy (BMBWF-V/3b/2023-0.188.679). Five-week-old C57BL/6J mice were fed a high-saturated fat and cholesterol diet (AIN-76 Western Diet, Test Diet, St. Louis, MO, USA) with water supplemented with sucrose/fructose (42 g/L) for 10 weeks. At 15 weeks, mice were treated intraperitoneally with either glycyrrhizin (NSC 167409, 6 mg/kg, Selleck Chemicals, Houston, TX, USA, n = 6) or compound control (12.5% DMSO, 10% PG, 1% Tween-20 in PBS, n = 6–9) once daily for four consecutive days. On the forth day of glycyrrhizin treatment LR was performed according to the Mitchell and Willenbrings 2/3 partial hepatectomy model and mice sacrificed 24 h or 48 h post-operation [21]. Plasma ALT and AST levels were measured using FUJI DRI-CHEM NX500 (Lab Technologies, Wien, Austria).

### 4.4. Immunofluorescence Staining

Cryo-embedded liver sections were permeabilized (0.1% Triton X-100) and incubated with primary antibodies (anti-rabbit HMGB1, 1:100, ab18256, Abcam, Cambridge, UK; anti-rat KI67, 1:100; anti-rabbit p21, 1:100, Invitrogen, Carlsbad, CA, USA; anti-cCasp3, CST, Danvers, USA) overnight at 4 °C and secondary antibodies (DyLight650 donkey anti-rabbit, 1:100, Abcam; DyLight 550 donkey anti-rat, Invitrogen) for 2 h at room temperature. Nuclei were counterstained using Hoechst 33342 and slides were mounted with ProlongGold Antifade reagent. Microscope images were obtained using Widefield Fluorescence Nikon A1plus Ti Microscope (Nikon, Vienna, Austria) and scale bars were added in ImageJ Fiji. Data analysis of the images was done using CellProfiler 3.1.9.

### 4.5. H&E and Oil Red O Staining

Liver sections were stained with hematoxylin and eosin (H&E) or oil red O following a standard protocol. Microscope images were captured using TissueGnostics slide scanner 7.1 (TissueGnostics GmbH. Vienna, Austria), and analysis was conducted using CellProfiler version 3.1.9.

### 4.6. RNA Extraction and Real-Time qPCR Analysis

Total RNA from harvested livers was isolated using TriFast™ reagent according to the manufacturer’s protocol (VWR, Radnor, PA, USA).Specific oligonucleotide primers were designed as follows: Hprt, fwd 5′-TCAGTCAACGGGGGACATAAA-3′, rev 5′-GGGGCTGTACTGCTTAACCAG-3′; KI-67, fwd 5′-AATCCAACTCAAGTAAACGGGG-3′, rev 5′-TTGGCTTGCTTCCATCCTCA-3′; Proliferating cell nucear antigen (PCNA), fwd 5′-GAACCTCACCAGCATGTCCA-3′, rev 5′-ATTCACCCGACGGCATCTTT-3′; Cyclin D1, fwd 5′-CTGGATGCTGGAGGTCTGTG-3′, rev 5′-TCATCCGCCTCTGGCATTTT-3′.

### 4.7. Statistical Evaluation

Statistical analyses were conducted using Graphpad Prism 8.0.2 (GraphPad Software Inc., San Diego, CA, USA) or IBM SPSS Statistics 20 (SPSS, Inc., Chicago, IL, USA). Gaussian distribution was assessed using Anderson Darling (A2*), D’Agostino-Pearson omnibus (K2), Shapiro-Wilk (W), and Kolmogorov-Smirnov (distance) tests. Time point comparisons were analyzed with a paired two-tailed *t*-test for Gaussian distributed data, and a two-tailed Wilcoxon matched-pairs signed-rank test for non-Gaussian distributed data. Group comparisons (PHLF vs. no PHLF) and time point differences were evaluated using a two-way ANOVA with Tukey correction for multiple comparisons. A significance level of *p* < 0.05 was considered statistically significant, and a 95% confidence interval was applied.

## Figures and Tables

**Figure 1 ijms-25-07150-f001:**
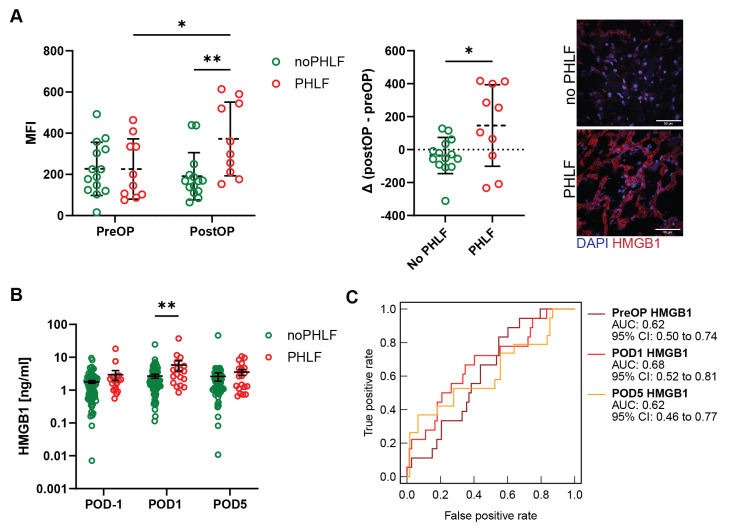
High mobility group box protein-1 (HMGB1) levels correlate with post-hepatectomy liver failure (PHLF) and liver injury markers ALT (alanine-aminotransferase) and AST (aspartate-aminotransferase). (**A**) Immunofluorescence staining and quantification of HMGB1 in liver biopsies before and after resection, with or without PHLF. (Two-way ANOVA: * *p* < 0.05, ** *p* < 0.01). Regeneration-induced effects depicted as difference after-before resection (∆ postOP-preOP) (unpaired *t*-test: * *p* < 0.05). (**B**) Plasma HMGB1 levels before and after partial hepatectomy from patients with and without PHLF (Two-way ANOVA: ** *p* < 0.01). (**C**) ROC (receiver operation curve) analysis comparing HMGB1 predictive potential with PHLF. (**D**) Correlation of plasma HMGB1 with AST and ALT levels on POD1.

**Figure 2 ijms-25-07150-f002:**
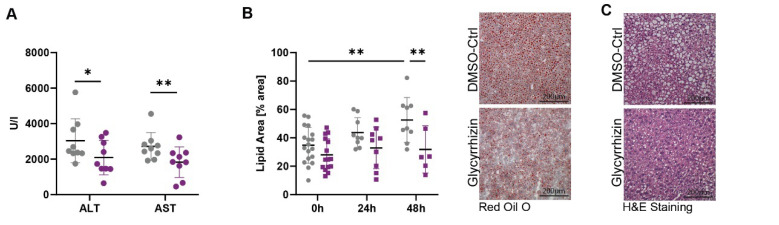
Inhibiting high mobility group box protein-1 (HMGB1) with glycyrrhizin enhances liver regeneration. (**A**) Alanine-aminotransferase (ALT) and aspartate-aminotransferase (AST) levels in 15-week-old mice on a fast food diet, treated with glycyrrhizin or DMSO control, 24 h post partial hepatectomy (PHx) (unpaired *t*-test: * *p* < 0.05, ** *p* < 0.01). (**B**,**C**) Oil Red O Staining and Hematoxylin & eosin (H&E), 48 h post PHx (Two-way ANOVA: ** *p* < 0.01). (**D**) Gene expression of KI67, PCNA, and Cyclin D1 before and 48 h post PHx (Two-way ANOVA: ** *p* < 0.01, *** *p* < 0.001, **** *p* < 0.0001). (**E**,**F**) Immunofluorescence quantifications of KI67 and cleaved caspase 3 (cCasp3) and p21 pre- and post-PHx (Two-way ANOVA: * *p* < 0.05, ** *p* < 0.01, *** *p* < 0.001, **** *p* < 0.0001).

**Figure 3 ijms-25-07150-f003:**
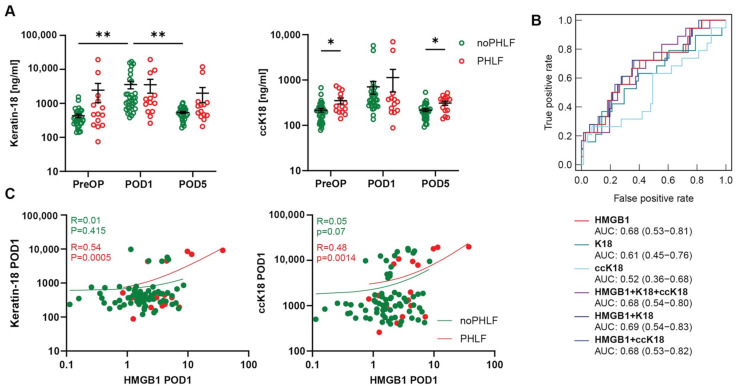
High mobility group box protein-1 (HMGB1) levels are associated with cell death marker keratin-18 (K18) and cleaved caspase cytokeratin-18 (ccK18). (**A**) Plasma Keratin-18 (K18) and cleaved caspase cytokeratin-18 (ccK18) levels pre- and post- partial hepatectomy (PHx) in patients with and without post-hepatectomy liver failure (PHLF) (Two-way ANOVA: * *p* < 0.05, ** *p* < 0.01). (**B**) Receiver operating characteristic (ROC) curve comparing HMGB1, K18 and ccK18 postoperative day 1 (POD1) with PHLF. (**C**) Correlation of plasma HMGB1 with K18 and ccK18 on POD1. ROC curves of HMGB1 were compared to ROC curves of PDR and R15 in a subset of 64 patients (**D**) and in a subset of 39 patients ROC curves of HMGB1 and APRI+ALBI were compared (**E**).

**Table 1 ijms-25-07150-t001:** Plasma cohort patient characteristics.

Parameter		Cohort (n = 96)	PHLF Cohort (n = 22)	no PHLF Cohort (n = 74)	Missing Values
Sex					
	Male	58 (60.4%)	13 (59.1%)	45 (60.8%)	
	Female	38 (39.6%)	9 (40.9%)	29 (39.2%)	
Age (years)		63.3 (23.0–86.1)	67.6 (35,0–82.4)	62.5 (23.0–86.1)	
Hepatic resection					
	Minor (<3 segments)	23 (24.0%)	2 (9.1%)	21 (28.4%)	
	Major (≥3 segments)	73 (76.0%)	20 (90.9%)	53 (71.6%)	
Tumor type					
	CRLM	39 (40.6%)	7 (31.8%)	32 (43.2%)	
	HCC	17 (17.7%)	5 (22.7%)	12 (16.2%)	
	CCA	18 (18.8%)	7 (31.8%)	11 (14.9%)	
	Pancreas related	10 (10.4%)	1 (4.5%)	9 (12.2%)	
	Others	12 (12.5%)	2 (9.1%)	10 (13.5%)	
Hepatic comorbidities					
	Steatosis (%)	15.0 (0.0–100.0%)	6.0 (0.0–40.0%)	17 (0.0–100.0%)	
	Steatohepatitis	44 (45.8%)	7 (31.8%)	37 (50.0%)	3 (3.1%)
	Fibrosis	40 (41.7%)	7 (31.8%)	33 (44.6%)	3 (3.1%)
	CASH	12 (12.5%)	4 (18.2%)	8 (10.8%)	2 (2.1%)
Treatment					
	Intraoperative RBCs transfusion	16 (16.7%)	7 (31.8%)	9 (11.8%)	
	Neoadjuvant CTx	38 (39.6%)	8 (36.4%)	30 (40.5%)	4 (4.2%)
Preoperative parameters					
	PDR (%)	20.99 (9.7–40.0)	20.0 (9.7–40.0)	21.3 (12.5–31-4)	27 (28.1%)
	Platelets (×10^3^/µL)	238.0 (92.0–465.0)	238.4 (129.0–345.0)	237.8 (92.0–465.0)	17 (17.7%)
	SB (mg/dL)	0.5 (0.1–6.6)	0.7 (0.2–6.6)	0.5 (0.1–2.3)	16 (16.7%)
	PT (%)	103.0 (64.0–150.0)	98.0 (45.0–150.0)	102.0 (45.0–150.0)	17 (17.7%)
	AP (U/L)	84.0 (30.0–707.0)	81.0 (51.0–707.0)	87.0 (30.0–254.0)	16 (16.7%)
	GGT (U/L)	52.0 (7.0–1576.0)	53.0 (18.0–1576.0)	49.0 (7.0–710.0)	16 (16.7%)
	AST (U/L)	30.5 (17.0–144.0)	34.0 (14.0–224.0)	32.0 (14.0–224.0)	33 (34.4%)
	ALT (U/L)	30.0 (7.0–143.0)	33.5 (12.0–129.0)	29.0 (7.0–143.0)	16 (16.7%)
	Albumin (g/L)	43.3 (33.0–243.0)	41.3 (33.0–47.3)	44.4 (33.7–243.0)	37 (38.5%)
Morbidity					
	No morbidity	50 (52.1%)	5 (22.7%)	45 (59.2%)	
	Grade I	8 (8.3%)	2 (9.1%)	6 (7.9%)	
	Grade II	21 (21.9%)	6 (27.3%)	15 (19.7%)	
	Grade III	10 (10.4%)	4 (18.2%)	6 (7.9%)	
	Grade IV	3 (3.1%)	2 (9.1%)	1 (1.3%)	
	Grade V	4 (4.2%)	3 (13.6%)	1 (1.3%)	
Postoperative stay					
	ICU (days)	2.0 (0.0–18.0)	3.0 (0.0–15.0)	1.0 (0.0–18.0)	
	Total hospitalization (days)	14.0 (3.0–117.0)	22.0 (5.0–117.0)	11.0 (3.0–44.0)	
PHLF ISGLS					
	no PHLF	76 (77.1%)		76 (100.0%)	
	Grade A	7 (7.3%)	7 (31.8%)		
	Grade B	6 (6.3%)	6 (27.3%)		
	Grade C	9 (9.4%)	9 (40.1%)		

ALT, alanine aminotransferase; AP, alkaline phosphatase; AST, aspartate aminotransferase; CASH, chemotherapy-induced acute steatohepatitis; CCA, cholangiocellular carcinoma; CTx, chemotherapy; GGT, gamma-glutamyl transpeptidase; HCC, hepatocellular carcinoma; ICU, intensive care unit; CRLM, metastases of colorectal carcinoma; PHLF, post-hepatectomy liver failure; PDR, plasma disappearance rate; PT, prothrombin time; RBCs, red blood cells; SB, serum bilirubin; bold: PHLF vs. no PHLF: *p* > 0.05.

**Table 2 ijms-25-07150-t002:** Staining cohort patient characteristics.

Parameter		Subcohort (*n* = 24)	PHLF Cohort (*n* = 10)	no PHLF Cohort (*n* = 14)	Missing Values
Sex					
	Male	9 (37.5%)	4 (40.0%)	5 (35.7%)	
	Female	15 (62.5%)	6 (60.0%)	9 (64.3%)	
Age (years)		59.1 (35.0–81.3)	60.2 (35.0–76.9)	58.3 (38.4–81.3)	
Hepatic resection					
	Minor (<3 segments)	0 (0.0%)	0 (0.0%)	0 (0.0%)	
	Major (≥3 segments)	24 (100.0%)	10 (100.0%)	14 (100.0%)	
Tumor type					
	CRLM	4 (16.7%)	0 (0.0%)	4 (28.6%)	
	HCC	2 (8.3%)	1 (10.0%)	1 (7.1%)	
	CCA	13 (54.2%)	8 (80.0%)	5 (35.7%)	
	Pancreas related	2 (8.3%)	1 (10.0%)	1 (7.1%)	
	Others	3 (12.5%)	0 (0.0%)	3 (21.4%)	
Hepatic comorbidities					
	Steatosis (%)	5.0 (0.0–25.0%)	2.0 (0.0–10.0%)	8.0 (0.0–25.0)	
	Steatohepatitis	7 (30.4%)	2 (20.0%)	5 (38.5%)	1 (4.2%)
	Fibrosis	10 (43.5%)	5 (50.0%)	5 (38.5%)	1 (4.2%)
	CASH	4 (17.4%)	1 (10.0%)	3 (23.1%)	1 (4.2%)
Treatment					
	Neoadjuvant CTx	6 (35.3%)	3 (42.7%)	3 (30.0%)	7 (29.2%)
	Intraoperative RBCs	3 (12.5%)	2 (20.0%)	1 (7.1%)	
Preoperative parameters					
	PDR (%)	25.0 (15.0–32.0)	26.0 (22.0–32.0)	24.5 (15.0–31.0	7 (29.2%)
	Platelets (×10^3^/µL)	283.0 (144.0–1082.0)	242.2 (172.0–333.0)	324.0 (144.0–1082.0)	4 (16.7%)
	SB (mg/dL)	1.3 (0.2–6.1)	2.0 (0.3–6.1)	0.6 (0.2–2.5)	4 (16.7%)
	PT (%)	104.0 (73.0–126.2)	95.3 (73.0–124.0)	113.0 (97.0–126.2)	6
	AP (U/L)	238.0 (42.0–1432.0)	468.3 (65.0–1432.0)	98.0 (42.0- 158.0)	4 (16.7%)
	GGT (U/L)	228.0 (7.0–2576.0)	335.4 (27.0–2576.0)	121.0 (7.0–382.0)	8 (33.3%)
	AST (U/L)	40.0 (14.0–64.0)	39.0 (14.0–65.0)	40.0 (19.0–64.0)	8 (33.3%)
	ALT (U/L)	49.0 (10.0–257.0)	61.0 (12.0–257.0)	39.0 (10.0–127.0)	6 (25.0%)
	Albumin (g/L)	43.1 (32.1–74.2)	37.9 (34.4–44.5)	46.6 (32.1–74.2)	7 (29.2%)
Morbidity					
	No morbidity	8 (33.3%)	1 (10.0%)	7 (50.0%)	
	Grade I	4 (16.7%)	2 (20.0%)	2 (14.3%)	
	Grade II	5 (20.8%)	2 (20.0%)	3 (21.4%)	
	Grade III	3 (12.5%)	1 (10.0%)	2 (14.3%)	
	Grade IV	2 (8.3%)	2 (20.0%)	0 (0.0%)	
	Grade V	2 (8.3%)	2 (20.0%)	0 (0.0%)	
Postoperative stay					
	ICU (days)	3.0 (0.0–15.0)	4.0 (0.0–15.0)	2.0 (0.0–9.0)	
	Total hospitalization (days)	11.0 (3.0–24.0)	12.0 (3.0–23.0)	11.0 (4.0–24.0)	
PHLF ISGLS					
	no PHLF	14 (58.3%)		14 (100%)	
	Grade A	3 (12.5%)	3 (30.0%)		
	Grade B	4 (16.7%)	4 (40.0%)		
	Grade C	3 (12.5%)	3 (30.0%)		

ALT, alanine aminotransferase; AP, alkaline phosphatase; AST, aspartate aminotransferase; CASH, chemotherapy-induced acute steatohepatitis; CCA, cholangiocellular carcinoma; CTx, chemotherapy; GGT, gamma-glutamyl transpeptidase; HCC, hepatocellular carcinoma; ICU, intensive care unit; CRLM, metastases of colorectal carcinoma; PHLF, post-hepatectomy liver failure; PDR, plasma disappearance rate; PT, prothrombin time; RBCs, red blood cells; SB, serum bilirubin; bold: PHLF vs. no PHLF: *p* > 0.05.

## Data Availability

Data is contained within the article or Appendix A.

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
