# Peer review of "HMGB1-Mediated Cell Death—A Crucial Element in Post-Hepatectomy Liver Failure"

_ijms, 2024, doi:10.3390/ijms25137150_

Round 1

Reviewer 1 Report

Comments and Suggestions for Authors

see attached

Comments on the Quality of English Language

English is mostly fine.

Author Response

Response to Reviewer 1:

We thank the reviewer for the careful evaluation of our manuscript and the excellent suggestions, which helped to improve the overall quality of the work.

Please find a point-to-point reply below:

  1. Abstract: I would suggest to format the abstract into Intro/Methods/Discussion, it makes it easier to read.

Reply: We agree with the reviewer that this makes it easier to read. However, the Author guidelines of IJMS state that headings should not be used: “Abstract: The abstract should be a total of about 200 words maximum. The abstract should be a single paragraph and should follow the style of structured abstracts, but without headings”. We have therefore not changed it, but are happy to do so if this is ok with the journal.

  1. Introduction: several methods to asses/predict PHLF have been established (LImax, APRI/ALBI…). They should be mentioned.

Reply: We thank the reviewer for this important comment and have added a paragraph on established PHLF risk assessment tools and also compared HMGB1 to these markers in Fig.3. See comment below.

  1. Is there any data on the predictive role of HMGB1?

Reply: To the best of our knowledge we are the first to show in human data on HMGB1 in the context of PHLF. In the revised manuscript we have now added comparisons of HMGB1 ROC curves with other risk assessment scores (ICG clearance and ABRI+ALBI) in Fig.3D and E. As stated in the manuscript not from all patients data were available. Therefore, the analysis could only be performed in a subset of patients. We therefore also state in the discussion, that these data have to be considered with care and validation studies are warranted.

  1. Methods: Mice were treated with glycyrrhizin before surgery for 4 days. When was surgery in this context?

Reply: Thanks for pointing out this lack in clarity. Mice were treated 72h, 48h, and 24h prior as well as immediately before PHx. We rephrased the methods section accordingly and there is also a graphical overview of the experimental set-up in Suppl. Fig. 1.

  1. Results: Was there any preoperative risk assessment regarding PHLF performed in the LR cohort? It would be interesting to correlate this to the biomarkers in this study.

Reply: We thank the reviewer for this excellent suggestion. As stated above we have added data in Fig.3.

  1. I would suggest to abbreviate metastases of colorectal carcinoma with “CRLM” and cholangiocellular carcinoma with “CCA”, as is common.

Reply: We have now introduced the abbreviations.

  1. As I understand, liver tissue samples were obtained intraoperatively from patients before or after disruption of portal vein blood flow. Fig 1 shows that even in this very early context, PHLF seems to be predictable using HMGB1? This should be discussed because in the clinical context, PHLF often shows days after is often linked to complication (e.g. biliary leakage/ surgical site infection or e.g. pneumonia). However, with this data PHLF, seems to be predictable because of liver-specific damage/alterations before operation? This could be explored in detail in Discussion.

Reply: We thank the reviewer for raising this important point and have added a paragraph discussing it to the discussion section.

  1. ccK18 was elevated even before operation in patients with liver failure and on d5 but not on d1. What is the putative mechanism? Preexisting liver damage?

Reply: This is an interesting question. We think it has to do with pre-existing liver damage. On d1 all patients, those with and those without PHLF show elevated levels due to surgery, thereby no differences can be observed. We have added this speculation also in the revised discussion of the manuscript.

  1. When was PHLF diagnosed in the cohort of patients? (Especially in the context of postoperative complications).

Reply: We thank the reviewer for pointing out this lack of information, we added a paragraph to clarify this to the methods section, which now states:

“Morbidity was defined using the criteria put forth by Dindo et al (Dindo, Demartines, & Clavien, 2004). Severe morbidity was classified as grade 3 or higher. PHLF was defined and graded according to the criteria established by the International Study Group of Liver Surgery. The development of PHLF was characterized by elevated serum bilirubin (SB) levels and prolonged prothrombin time (PT) persisting on the fifth postoperative day. If SB and PT were already abnormal before surgery, SB had to be higher and PT lower than the preoperative values. Patients who were excluded from routine postoperative blood draws due to good clinical performance or early discharge were categorized as not having PHLF. To accurately represent the proportion of patients with clinically significant, symptomatic PHLF, it was classified as PHLF grades B and C (PHLF B+C), while the absence of PHLF was indicated by no PHLF or PHLF grade A.”

  1. The authors state, that no correlation between HMBG1 and neutrophil recruitment was found (l240) I fail to see this data in the study?

Reply: We now added a statement: “We further analyzed correlations between HMGB1 and markers of neutrophil activation, including neutrophil elastase (NE), myeloperoxidase (MPO), and citrullinated histone 3 (citH3), but found no associations (data not shown).” Since all data are negative we did not include the actual graphs in the manuscript.

  1. Discussion could be more detailed regarding predictive and therapeutic option. Also, if HMGB1 has a “crucial role” in PHLF is certainly still debatable at this point.

Reply: We have added more details regarding predictive and therapeutic option in the manuscript.

Reviewer 2 Report

Comments and Suggestions for Authors

The authors evaluated the association of the DAMP molecule HMGB1 with the post-hepatectomy liver failure in patients who underwent liver resection for various liver tumors. Additionally, they assessed the potential effect of glycyrrhizin (HMGB1 inhibitor) in murine models. The idea is novel and interesting, the methods are described in detail. References are fine.

There are, however, several points to consider:

Line 81: ‘All patients were Child-Pugh class A without encephalopathy, with compensated liver function’- This statement presumes that all of the patients had liver cirrhosis, since the CTP classification is used only in patients with LC.

Line 87 ‘Plasma was prepared as described.’- described where? According to the reference? Please change the expression.

Table 1 and 2:  intraoperative RBC transfusion is not a hepatic comorbidity; morbidity grades I-V- according to which classification?

Line 164-166: The authors should mention AUROC values in the text since these information are important regarding the performance of HMGB1 as a predictive marker for PHLF.

Line 166: Correlation coefficients of 0.55 and 0.62 are rather moderate than strong.

Lines 204 and 205: AUROC values of 0.52 and 0.61 are not high. Please interpret the results properly.

The authors should not overinterpret the study results (strong, high), but maintain an objective view.

They should also mention the most important statistical results (e.g. correlation coefficients, AUROC values) in text, not only in Figures where they are scarcely visible.

Comments on the Quality of English Language

The English language is fine. 

Author Response

Response to Reviewer 2:

We thank the reviewer for the careful evaluation of our manuscript and the excellent suggestions, which helped to improve the overall quality of the work.

Please find a point-to-point reply below:

The authors evaluated the association of the DAMP molecule HMGB1 with the post-hepatectomy liver failure in patients who underwent liver resection for various liver tumors. Additionally, they assessed the potential effect of glycyrrhizin (HMGB1 inhibitor) in murine models. The idea is novel and interesting, the methods are described in detail. References are fine.

There are, however, several points to consider:

  1. Line 81: ‘All patients were Child-Pugh class A without encephalopathy, with compensated liver function’- This statement presumes that all of the patients had liver cirrhosis, since the CTP classification is used only in patients with LC.

Reply: We thank the reviewer for pointing out this lack of clarity and now changed the wording to “all patients with liver cirrhosis…”.

  1. Line 87 ‘Plasma was prepared as described.’- described where? According to the reference? Please change the expression.

Reply: We thank the reviewer for pointing out that the wrong reference has been used. We deleted the sentence and now describe the plasma isolation process. It now reads: “Whole blood samples were collected into pre-chilled CTAD (citrate, theophylline, adenosine, and dipyridamole) tubes, immediately placed on ice, and processed within 30 minutes. Initially, the samples were centrifuged at 1000 x g at 4°C for 10 minutes. The supernatants were then centrifuged at 10,000 x g at 4°C for another 10 minutes to remove any remaining platelets. Plasma samples were aliquoted and stored at -80°C until analysis.”

  1. Table 1 and 2:  intraoperative RBC transfusion is not a hepatic comorbidity; morbidity grades I-V- according to which classification?

Reply: We thank the reviewer for pointing this out, we now moved RBC transfusion and CTx treatment to the “treatment section”. Morbidity was defined using the criteria put forth by Dindo et al (https://pubmed.ncbi.nlm.nih.gov/15273542/). We have added this now in the manuscript.

  1. Line 164-166: The authors should mention AUROC values in the text since this information are important regarding the performance of HMGB1 as a predictive marker for PHLF.

Reply: We agree with the reviewer and have added the information to the results section. Now line 188.

  1. Line 166: Correlation coefficients of 0.55 and 0.62 are rather moderate than strong.

Reply: We thank the reviewer for pointing this out and have revised the statement.

  1. Lines 204 and 205: AUROC values of 0.52 and 0.61 are not high. Please interpret the results properly.

Reply: We thank the reviewer for this comment and have revised the statement.

  1. The authors should not overinterpret the study results (strong, high), but maintain an objective view. They should also mention the most important statistical results (e.g. correlation coefficients, AUROC values) in text, not only in Figures where they are scarcely visible.

Reply: We have changed this throughout the manuscript.